# The Importance of Environmental Food Quality Labels for Regional Producers: A Slovak Case Study

**DOI:** 10.3390/foods11071013

**Published:** 2022-03-30

**Authors:** Jana Jaďuďová, Ján Tomaškin, Janka Ševčíková, Peter Andráš, Marek Drimal

**Affiliations:** Department of Environmental Management, Faculty of Natural Sciences, Matej Bel University in Banská Bystrica, Tajovského 40, 974 01 Banská Bystrica, Slovakia; jan.tomaskin@umb.sk (J.T.); janka.sevcikova@umb.sk (J.Š.); peter.andras@umb.sk (P.A.); marek.drimal@umb.sk (M.D.)

**Keywords:** food quality labels, regional producer, food products, Slovak Republic

## Abstract

Regional products are considered an important element of European cultural identity, contributing to the development and sustainability of rural areas. The article presents the research of regional labels from the territory of the Slovak Republic. Our research was aimed at determining the views of producers towards the regional product labeling scheme. The survey took place from January 2020 to April 2020 using an online questionnaire via Google Docs. The results obtained from the questionnaire survey were statistically processed: *t*-test, Fisher’s exact test, and Pearson correlation coefficient. We used Microsoft Excel and IBM SPSS Statistics 23 software. Based on the research results, we can state that two-thirds of producers (71.8%) are aware of the concept of a regional product. Most producers (82.0%) associate this concept with tradition and a specific region. They consider regional product labeling to be a tool to support the development of tourism (36.0%). A quarter of producers (25.7%) joined this scheme in order to add value to their products and help make consumers perceive them as safer products of higher quality. Based on the research results, we would recommend increasing the promotion of regional products on the part of the label coordinator.

## 1. Introduction

Regional products are considered an important element of European cultural identity, contributing to the development and sustainability of rural areas. The European Union has been involved in regional product labeling since the early 1990s. Regional product labeling is part of a broader product labeling concept that dates back to 1978 in Germany [1]. Regional product labeling is one of the options for the sustainable development of a certain region, as it is aimed at local entrepreneurs, and thus at supporting the social, cultural, and environmental development of the economy. According to Sharma [2], the local economy can benefit from the production and sale of local products, as less money ends up with national and multinational companies. A regional product label develops agriculture and local business, initiates the creation of new jobs, increases the volume of taxes and reinvestment in the region, and strengthens tourism. Jaďuďová, Rezníček [3] identified the theoretical and practical benefits of regional product labeling for rural development. According to them, the theoretical benefits include the satisfaction of the local community and producers of regional products with the quality of their living standards. The practical benefits include the possibility of easier access to the local labor market and an increase in the economic level of the region.

For consumers, knowledge about food is of basic interest, and information about products plays an important role in serving it. However, the ability of consumers to perceive the status of certain product characteristics may be limited as might be the case regarding the geographical origin, methods of production, and the regionality of products [4]. The definition of a regional product, as it is understood by experts in the field and consumers, varies. A regional product can be a product (souvenir), its production, place, accommodation, idea, organization, food, or drink. Most commonly associated with the term “regional product” are food and drinks. Enteleca Research and Consultancy [5] considers a regional product to be a local food or drink that is produced or grown in a given locality or is considered a specialty with a local name. These are mostly specific farm products, products with a regional label, and specialties unique to a given region. There are two perspectives on the term “local”; it is either that production uses exclusively local produce or also imported produce that is processed by local producers. If the locality aspect is to be assessed, it can be said that producers place great importance on the origin of the product and the production process, while the average consumer is more sensitive to the image-related side of local products, such as the product names or origin, [6]. Sodano [7] considers the following to be the specifics of a regional product:Geographic specificity. The product must have intrinsic characteristics that differentiate it from similar products. These characteristics must depend on a specific aspect of the production process, and/or in the raw agricultural input, that can be found only in a well-defined geographic area.Historical tradition. There must be historical evidence of the existence of the product in the past, with characteristics similar to the present.Cultural and social specificity. In the region where the product comes from, there is a consensus, depending on the local social and cultural environment, about the identification and appraisal of specific attributes that differentiate the product. The cultural value can be associated with particular celebrations or local gastronomic customs, or a symbology supporting local social norms.

Regional products can be understood as products associated with a particular, relatively bordered territory, with the region of their origin, constituting a deliberate component of their quality [8]. In the literature, we find a series of studies of positive features of the regional products, such as: a product’s local character links the product to sustainable consumption, support for local economies, or the proximity between production and consumption [9,10]; its regional character gives the product superior quality [11]; and its traditional character causes consumers to perceive these products as simple, natural and pure [12].

In Western Europe and North America, the interest in regional products began to grow in the early 1990s [8]. In Slovakia, it is a relatively new phenomenon. The existing Slovak studies deal primarily with the perception of consumers [1,13,14,15,16,17,18]. Hrubalova’s research [19,20] focuses on the regional product labeling scheme in the context of tourism. We found studies about the regional product in the context of regional development, such as by Štensová [21]. The research gap in the regional label studies is the producers ‘views on this labeling scheme. Little attention has also been paid to the significance of regional crafts products from the perspective of producers in Kotradyová, Borysk [22]. In our research, we focused on farm products and farmers. We find a similar study focused on producer research in the Czech Republic. Sadílek [4] realized the research in the two regions of the Czech Republic (Bohemia and Moravia). He analyzed the perception of food producers regarding the labeling scheme. Our research maps the producers’ views in one region of Slovakia because this theme is missed in the literature.

Slovakia is currently trying to create a unified product labeling system. In charge of the coordination and promotion of regional labels is the National Network of Local Action Groups of the Slovak Republic (NSS LAG), which brings together 17 local action groups, 11 private-public partnerships, and civic associations. The role of the local action group coordinating a regional label is to promote, support, and raise the profile of the regional label [16]. Each regional label has its own website. Their logo has a uniform appearance and each expresses a typical symbol of the region. The following principles have been adopted for the certification of regional products: the product must come from the region (be local), must be uniquely associated with the region, and must exhibit a certain share and tradition of handicraft and environmental friendliness [23]. Quality labels may build on graphics or other symbols that can be placed on a product or its packaging indicating that the product or the production process complies with certain standards and that this compliance has been certified [24,25,26]. Such standards may include classical product quality requirements but also criteria referring to the production process, country or region of origin, the composition of the product, health benefits of the product, etc. [27].

According to the available information, there are 12 active regional labels in Slovakia (Figure 1). Others are under preparation, such as: Podunajsko regional product, Tatry-Spiš regional product, Modra-Small Carpathians Wine Route regional product, and Liptov regional product.

We agree with the statement of [4] that quality labels have become a central component of consumer policy. They are a valuable tool for managing and communicating a higher quality and safety of food products for gaining a competitive advantage in the market. The importance of quality labels and the information they communicate have increased due to crises and scandals (e.g., BSE–Bovine Spongiform Encephalopathy) which have shaken the European food market over the past years, leading to a decline in consumer confidence in the safety and quality of food products.

## 2. Materials and Methods

The area of the research was the Hont region as defined by the territorial scope of the regional label (districts of Banská Štiavnica, Levice, Nové Zámky, Krupina, Veľký Krtíš, Žiar nad Hronom, Žarnovica, Figure 2). The subject of the research consisted of the producers of the Hont regional label. As the regional producers were located only in some of the districts, we narrowed the research sample to the districts of Banská Štiavnica, Krupina, Veľký Krtíš, and Levice.

The territory of the Hont region covers the southern part of central Slovakia, where Hont County used to be located. The borders of Hont County were formed by the Štiavnica Mountains to the north, the river Hron to the west, the river Krtíš to the east, and the rivers Danube and Ipeľ to the south. Ipeľ currently forms a natural border between the much larger Slovak and Hungarian parts of Hont. The centers of this region are the historic towns of Krupina and Banská Štiavnica. Characteristic elements of the region are mountainous settlements and the persistence of traditional folk customs. In some villages, traditional crafts such as lace-making, pottery, and blacksmithing have been preserved to this day. These traditions were popularised mainly by cultural events organized in several towns and villages [28]. The first Hont regional label was awarded in 2013.

The basic sample included 52 producers, this is all producers with the regional label Hont (Table 1). The selected sample was 39 producers, which completed the questionnaire. The people who participated in the survey learned about the study and the respondents, by filling out the questionnaire, agreed to publish it. The representativeness of the group of respondents depended on the gender and the geographical location of the individual regions in Slovakia (Table 2). In order to verify the representativeness of the sample, we used the nonparametric chi-squared test (χ^2^–test) whose principle consists in verifying the compliance of the expected theoretical distribution with the (empirical) distribution. According to the results of the test, we can state that the sample is representative by gender (*p*-value = 0.524) and by region (*p*-value = 0.957).

In the research of regional labeling, we analyzed the perception of regional labels among consumers. There is no information about the producers’ views of the label or their perception of the benefits or problems associated with its use. For this reason, our research was aimed at determining the views of producers towards the regional product labeling scheme. It included a summary modeling of the regional producer using decision-making factors such as knowledge of regional labeling, reasons for involvement in the regional labeling scheme, and regional label preference. The survey took place from January 2020 to April 2020 using an online questionnaire via Google Docs approaching producers in the Hont region who had a regional label for at least one of their products. Respondents were divided by gender, age, education, economic activity, monthly income, family status, the locality where they lived, and territorial classification. The other questions were focused on regional labeling. In the article were analyzed questions: 1. Assess the claims about your reasons for joining the regional labeling scheme: make myself visible, sales support, presentation of a product’s quality, prestige, invitation by the organizers of a contest, recommendation of friends, possibility of gaining a positive ranking for grants; 2. Indicate the logo you have noticed when shopping for foods: Regional product HONT, Quality from our regions, non-existent logo Regional product; 3. Which statement associates you with the term regional products: product based on handwork, a product that follows the customs and traditions of the region, a product produced only in the region, I don’t know; 4. How important do you think product labeling is in the region: support for employment and entrepreneurship in the region, tourism promotion, support for the region’s sustainability and development, the product is better than other products on the market, none; 5. Which product group do you prefer from regional products: regional food, meal, drink, craft product, organized event, course, workshop, regional accommodation or catering, none; 6. Your sales increased after joining the regional labeling scheme: yes, no, I am not able to evaluate it. According to the manual on the creation of a regional label of products and services for the Hont region, the Hont regional product label can be awarded to handicraft products, food and agricultural products, natural products, accommodation services, catering services, and events (Table 1).

The results obtained from the questionnaire survey were statistically processed. We used the following methods: Microsoft Excel, version 2016, Microsoft Corporattion, Washington and IBM SPSS Statistics software, version 23, IBM New Yourk. We established hypotheses: H1: We expect that the producers from the towns pay more attention to food labels; H2: We expect that the producers have seen an increase in sales since joining the regional labeling scheme. We used the methods of the *t*-test, Fisher’s exact test, and Pearson correlation coefficient. We tested at a significance level of 0.05, as the selected sample was small. Using the *t*-test we determined how the respondents evaluated arguments of involvement in a regional labeling scheme. The respondents rated the statements based on a five-point scale (1–totally agree, 5–totally disagree). Using Fisher’s exact test and Pearson correlation coefficient, we determined the degree of the dependence coefficient which determined the dependence between gender, age, education, locality, and selected factors on consumer behavior. The coefficient had values in the interval from <0,1> or <1,1>. A value of 0 indicated independence [30].

## 3. Results and Discussion

Women and men were equally represented in the survey (49.0% women, 51.0% men). The age range was generated based on the age limit for engaging in business; the current legislation in the Slovak Republic is one of the strictest in Europe and stipulates that workers must be 18 years or older. In our survey, producers in the productive age, i.e., in the age range of 26–61 years old predominated (62.5%). Younger producers in the age range of 18–25 years old (25.0%) and producers of retirement age, i.e., over 62 years old (22.5%), were equally represented. Younger producers considered regional labeling to be an aspect of starting up their business, and for producers of retirement age, the production of regional products is a lifelong activity. Based on the age structure of the producers, we can state that this was an unrepresentative sample. In terms of education, producers with a secondary education with “maturita” (school-leaving exam) predominated (53.8%). They were followed by producers with a first or second-degree university education (36.0%). Producers with a tertiary degree university education or higher comprised the lowest share among the group of producers (10.2%). No producer had primary education only. We state that the sample was not representative in terms of education.

The analysis of the reasons for the involvement of producers in the regional labeling scheme was evaluated by means of a *t*-test. The producers evaluated the submitted statements that were previously identified as the most common in scientific articles, using a five-point scale (1–totally agree, 5–totally disagree), as shown in Table 3. The significance level was lower than α = 0.05 for all of the statements. Producers rated almost all statements very similarly, with a variance of 2.18–2.79 (to make myself visible, for sales support, presentation of a product’s quality, prestige, recommendation of friends). However, the values for the statement “by the organizers of a contest” were significantly different (mean = 4.82, *t*-value = 65.996). Based on the producers’ replies, we could conclude that the dominant response to this statement was “totally disagree”. In the Hont region, entry into the regional labeling scheme does not take place at the invitation of the label coordinator. Sadílek [4] found very similar values for all statements when researching producers in the Czech Republic and Moravia (mean between 2.00 and 2.92). The producers rather agreed with all the statements without a significant difference in their responses. In our research, we noted significant differences in the responses to “invitation by the organizers of a contest” (mean = 4.82) and “possibility of gaining a positive ranking for grants” (mean = 3.41).

The awareness of a regional label is a key characteristic for modeling a regional producer. A positive finding in our survey was that approximately two-thirds of the producers were aware of the concept of a regional product (71.8%). The regional product labels in the Slovak Republic managed by local action groups are characterized by a unified visual logotype style. They respected the common rules for awarding labels, which emphasized not only the regional origin of products but also their environmental friendliness and regional uniqueness. They considered the link to the tradition of the given region, the share of local produce, the share of handiwork, environmental friendliness, and the uniqueness of the products have been demonstrated [16]. A negative finding was that 5.1% of the producers indicated the possibility of a fictional logo (logo 3). Chalupová, Prokop, Rojík [31] tested which actual logo the respondents could identify in the Vysočina Region of the Czech Republic. A non-existent logo was recognized by almost a quarter of all respondents (23.4%). These producers likely did not pay attention to product labels, which was counterproductive, as they produced a regional product and had their own label (Table 4). We agreed with this view.

Dependence of the awareness of the Hont regional label and the respondents’ residence in a town/village was tested (Hypothesis H1). The results did not show a dependence between the monitored parameters (X^2^ = 2.212, sig. *p* = 0.331, *p* > 0.05). Table 5 shows that the producers living in villages were more familiar with the Hont regional logo (60.7%). When buying, producers living in cities paid more attention to logo 2 (Quality From Our Regions), which was determined by the availability of the products in multiple stores. The made-up logo 3 was indicated only by rural producers. We rejected Hypothesis H1. They were men of a productive age with secondary education and a monthly income of up to EUR 550. We can state that they buy food products sporadically, as they can provide them on their own. Chalupová, Prokop, Rojík [31] came to a different finding in their research in the Vysočina Region (Czech Republic), where 71.58% of the respondents who indicated the made-up logo were from cities and 67.0% of the respondents who were aware of the Vysočina regional product logo were from the countryside. Table 5 shows that the city dwellers were more aware of the existence of the labels, but they do not pay attention to them.

More than two-thirds of the respondents in the questionnaire survey (82.0%) associated a regional product with traditions and a specific region. Producers for whom the regional product was important in connection with customs and traditions slightly prevailed (46.0%). They were followed by producers for whom the product must be produced only in a certain region (36.0%), while only 18.0% of the producers considered handiwork to be the basis. Topca, Kaplan [32] also state a high level of ethnocentrism related to the reluctance to buy foreign products. The results of Fernández-Ferrín et al. [33] indicated that Galician producers had a significant influence to buy products from the region. Lang et al. [34] present a survey in the American continent. Some producers indicated that the uniqueness of the product to the region or the region’s notable characteristics were among the three most important reasons to buy a product, and 17 percent sought traditional foods or recipes from the region. Bianchi and Mortimer [35] found that the most influential factor in the consumption of local food among a sample of Australian and Chilean producers was their attitude toward local products, which, in turn, was affected by their favorable attitude toward supporting local businesses.

By means of further analyses, we examined the influence of identification factors (gender, age, education, location) in connection with their entry into the scheme of regional product labeling (Table 6). We came to the conclusion that the parameters we examined did not have a significant impact. The effect of gender was demonstrated in the importance of regional labeling (*p*-value = 0.014, Cramer’s V = 0.525). Men attached more importance to the development of tourism in connection with regional labeling, and women saw the importance of product labeling in promoting employment and entrepreneurship in the region. Atkin et al. [36] have a positive view of the connection between the agri-food sector in America and the development of tourism in the region, as the presence of traditional products has become a reason why tourists decide to visit the region. Another view was demonstrated by Hrubalova [19] in her research from the Slovak Republic, i.e., there are no regional product brands from regions with significant potential for tourism development. Based on the producers’ responses, we are inclined to agree with the statement of Atkin et al. [36].

Based on the results of Fisher’s exact test, we can state that age was not a determining factor (Table 6). We confirmed the effect of age only in connection with the preferred groups of regional products (*p*-value = 0.071, Cramer’s V = 0.464). With increasing age, people prefer regional food, meals, or drinks rather than accommodation or catering facilities, or an organized event, course, or workshop. As for education, we found a moderate dependence for the importance of regional labelling (*p*-value = 0.023, Cramer’s V = 0.480). Producers with higher levels of education (first, second, and third-degree university or higher education) did not associate regional products with tourism development or local business support but considered regional products to be safer and of a higher quality than other products on the market. These findings are supported by several studies focused on the demographic characteristics of consumers [37,38,39,40], which identified the so-called green consumers: women, younger people with a university degree, and a higher monthly income. Our results are consistent with these studies.

The last parameter examined was the location of the producers’ residence (town–countryside). A positive moderate dependence was confirmed for the importance of regional labeling (*p*-value = 0.009, Cramer’s V = 0.557). Producers from the countryside associated regional products with the support of local businesses. Zenetti, Klapper [41] state that consumers are less dependent on advertising if they have sufficient experience with the label or producers. A similar finding was arrived at by Soroka, Wojciechovska-Solis [42], as the inhabitants of eastern Poland and western Ukraine most often obtained information on regional food from their friends or family. Policy-makers in the Czech Republic have already recognized the importance of regional food products as a means to support the identity and economy of the region [25]. We agree with this view, but in Slovakia, there is small support for the government, which the producer also stated in the questionnaire.

We assumed that acquiring a regional label has helped producers to increase their sales (Hypothesis H2). By joining the organized scheme of regional product labeling, they obtained wider sales opportunities (yard sales, organized events, smaller shops with local products). Our assumption was based on a claim by Fernández-Ferrín et al. [33] stating that these products are sold through short chains, i.e., yard sales, which guarantees the origin of the food and method of production. Maeve, McIntyre [43] also emphasize the importance of direct sales, especially for regional foods. In our analysis, we found differences across the territorial classification of producers (Table 7). We did not prove a direct correlation between the increase in sales and the territorial classification of producers (*p*-value = 0.838, *r* = 0.034). We rejected Hypothesis H2. The surveyed districts (Banská Štiavnica, Krupina, Veľký Krtíš, and Levice) are among the districts with notoriously insufficient job creation, slow economic growth, and high unemployment rate. The sales of regional products are therefore the main source of income for two-thirds of the producers contacted. The district of Banská Štiavnica is dominated by producers who have not recorded an increase in sales after joining the regional product labeling scheme (67.0%). In the district of Veľký Krtíš, we found 56.0% of producers whose sales had increased (mainly local winemakers). The districts of Levice and Krupina have an even representation of producers whose sales increased and those who could not answer this question (Table 7). We did not confirm the view of Hu et al. [44] that consumers are willing to pay extra money for a regional product, thereby supporting small family farmers and craftsmen. Maeve, McIntyre [43] state that consumers in Ireland, Finland, France, Scotland, Spain, and Wales associate high prices with quality, for which they are willing to pay more.

Based on the analyses performed, we came to the conclusion that regional producers are aware of the regional product labeling scheme. The reason they join the product labeling scheme is the possibility to differentiate their products from other similar products, popularizing themselves and gaining recognition for their work in the process. They perceive a regional label as a sign of greater quality that will ensure the safety of their products. They did not notice a significant increase in the sales of their products after being awarded a regional label. Each producer would welcome more support from the state to promote regional products.

## 4. Conclusions

Health safety, sustainability, authenticity, or safety are product characteristics that an average consumer cannot verify and therefore expects from a trusted seller. The consumer cannot judge whether the product contains the ingredient listed on the packaging and whether or not the ingredient actually has the health effects claimed or whether the product was actually produced and processed according to organic production rules [45,46,47]. The consumer must therefore trust the producer and the farmer. The category of trusted producers includes regional producers, who were the center of focus of our article. We agree with Vandana, Verma, Sabharwal [48] in that the term ‘traditional (regional) product’ refers to food products that have been used for centuries and have been passed from one generation to another. They suppose that such food products in their original form have not been affected by modern technology, processing, or packaging. They are an expression of culture. These are the basic attributes of any regional product with an ambition to become a regional label.

A regional label is intended to support people living in the regions, revitalize rural development, and create motivation for people to stay in these regions. These are usually tradesmen, farmers, folk artists, collectors of medicinal herbs, or small family businesses, which are often far from urban settlements, where there is a low frequency of local traffic [49]. They produce so-called exceptional products, but they have problems selling them. They lack market knowledge, contacts, or sufficient funds for marketing communication to compete with cheaper alternatives from multinational companies.

Based on the research results, we can state that two-thirds of producers (71.8%) are aware of the concept of a regional product. In the questions asked via the questionnaire, they were able to identify the logo of the Hont regional product. Most producers (82.0%) associate this concept with tradition and a specific region. They consider regional product labeling to be a tool to support the development of tourism (36.0%). A quarter of producers (25.7%) joined this scheme in order to add value to their products and help make consumers perceive them as safer products of higher quality. However, joining the scheme did not bring them higher profits. On the contrary, they had a negative opinion of the cooperation with the label coordinator. Based on the research results, we would recommend increasing the promotion of regional products on the part of the label coordinator. The consumer does not buy a product if he/she does not know that it exists and that it bears a label guaranteeing a certain composition and origin. In order to increase promotion, it is also necessary to streamline cooperation between producers and the label coordinator, especially the transfer of information across the parties involved.

The authors of the study are aware of some limitations. First, the analysis was carried out in only one European country and should therefore be replicated with other brands to provide more proof. It is reasonable to monitor and compare countries that differ in their cultural environment. Second, we only looked at one method (questionnaire). Further research could examine interviews, allowing the authors to obtain additional information on their reasons for joining the regional labeling scheme.

## Figures and Tables

**Figure 1 foods-11-01013-f001:**
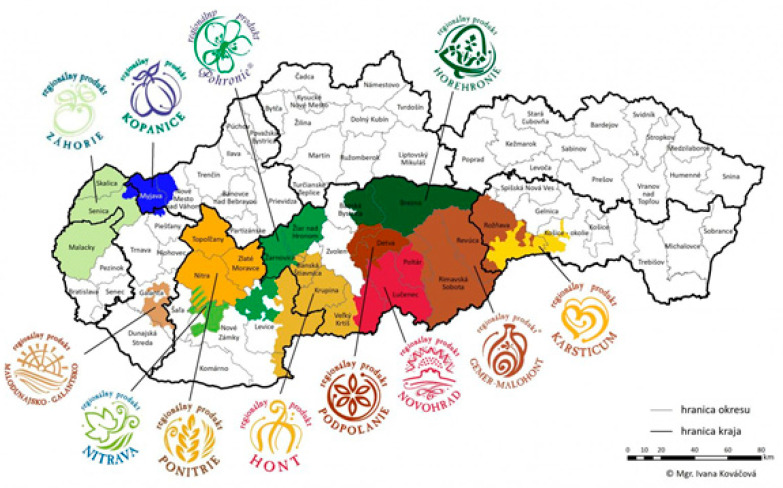
Established regional product labels in Slovakia. Adapted from [28].

**Figure 2 foods-11-01013-f002:**
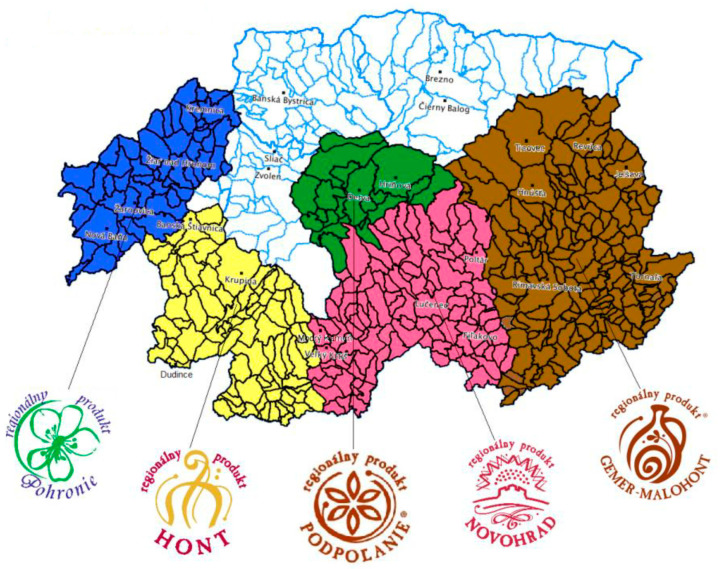
Demarcation of territory regional brands within the Banská Bystrica self-governing Region. Adapted from [28].

**Table 1 foods-11-01013-t001:** Division of certified regional products by product category.

Category	Subcategory	Number of Certified Products	% Share of All Certified Products
Craft products	wood, wicker, stone	6	38.5
	ceramics, clay glass, metal, wire textile, wool, lace animal skin other	6 2 2 1 3	
Food and agricultural products	bakery and confectionery products garden products farm products meat and sausages drinks other	5 5 2 0 4 1	32.7
Natural products	flowers and herbs mushrooms hones and honey products other	1 0 10 1	23.1
Accommodation services	hotel pension hostel cottage camp	0 1 0 0 0	1.9
Food services	food services	2	3.8
Events	events	0	0.0
Total		52	100.0

**Table 2 foods-11-01013-t002:** Relative primary and sample population by gender and district. Adapted from [29].

		Gender		District				
		Male	Female	Banská Štiavnica	Krupina	Veľký Krtíš	Levice	Total
Basic sample	absolute frequency	28	24	16	14	13	9	52
	relative frequency (%)	53.8	46.2	30.8	26.9	25.0	17.3	100.0
Selected sample	absolute frequency	19	20	12	10	9	8	39
	relative frequency (%)	51.3	48.7	30.8	25.6	23.1	20.5	100.0

**Table 3 foods-11-01013-t003:** *T*-test for statements on reasons to obtain a HONT regional label (*n* = 39).

	Totally Agree	Rather Agree	Neither Agree nor Disagree	Rather Disagree	Totally Disagree	Mean	Sig.	*T*-Value
Make myself visible	14	8	10	4	3	2.33	0.000	11.104
Sales support	10	6	10	8	5	2.79	0.000	12.422
Presentation of a product’s quality	15	10	8	4	2	2.18	0.000	10.980
Prestige	13	10	11	3	2	2.26	0.000	11.843
Invitation by the organizers of a contest	0	0	1	5	33	4.82	0.000	65.996
Recommendation of friends	6	18	15	0	0	2.23	0.000	19.299
Possibility of gaining a positive ranking for grants	3	6	12	8	10	3.41	0.000	16.779

**Table 4 foods-11-01013-t004:** Label characteristics in the survey.

Logo	Characteristic
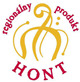	The HONT regional product label was established in 2012; the first local producers of agricultural products, food, and handicraft products received their certificates from the regional certification commission in 2013.
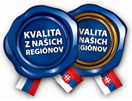	The “QUALITY FROM OUR REGIONS” project serves to support the sale of Slovak products and services. It is a social responsibility project based on consumer education through media campaigns and consumer competitions. The aim is to support the purchase of Slovak food, food products, and the use of domestic services.
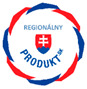	Non-existent logo, created by the authors.

**Table 5 foods-11-01013-t005:** Awareness of the regional labels in Hont depending on residence in a town or village.

	Residence: Town/Village
Awareness of the Label, Row rel. Frequencies (%) *	Town	Village	Total
Logo 1 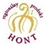	39.3	60.7	100.0
Logo 2 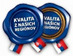	55.6	44.4	100.0
Logo 3 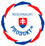	0.0	100.0	100.0
Pearson chi-sq. test	X^2^ = 2.212	df = 2	*p* = 0.331

* awareness of the label (answer Yes).

**Table 6 foods-11-01013-t006:** Dependence of identification factors on regional labeling.

		Preferred Product Group	Importance of the Regional Label	Attributes of the Regional Product
Gender	*p*-value Cramer’s V	0.831 0.201	0.014 0.525	0.764 0.127
Age	*p*-value Cramer’s V	0.071 0.464	0.860 0.241	0.490 0.189
Education	*p*-value Cramer’s V	0.394 0.261	0.023 0.480	0.617 0.186
Locality	*p*-value Cramer’s V	0.858 0.204	0.009 0.557	0.341 0.259

**Table 7 foods-11-01013-t007:** Impact of the territorial classification of producers on sales increases.

	Number of Words			
	Banská Štiavnica (*n* = 12)	Krupina (*n* = 10)	Veľký Krtíš (*n* = 9)	Levice (*n* = 8)
Yes	2	4	5	3
No	8	1	3	2
I am not able to evaluate it	2	5	1	3
Pearson correlation coefficient	*p*-value = 0.838		*r* = 0.034	

## Data Availability

The datasets used and/or analyzed during the current study are available from the corresponding author on reasonable request.

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
