# Peer review of "The Importance of Environmental Food Quality Labels for Regional Producers: A Slovak Case Study"

_foods, 2022, doi:10.3390/foods11071013_

Round 1

Reviewer 1 Report

The paper ‘The Importance of Environmental Food Quality Labels for Regional Producers: a Slovak Case Study’ is an evaluation of regional product perception in Slovakia. Authors try to summarize current situation the country and focused on identification main factors driving regional product and labelling process and development. The paper is interesting for the potential reader, but needs some improvements.

Comments:

The introduction should be extended by the analysis of the state of the art in available literature.

The discussion should be seriously extended and the extension, between others should be enriched as follows

  1. Please discuss all questions and factors identified within the questionnaire
  2. Although authors introduced some comments about results obtained in other countries, there is a need to add more detailed discussion regarding similar research in countries from the same region (east and central Europe) and other regions like West Europe, other continents.
  3. If results are different than presented in literature, please explain the reason of the differences

The questionnaire should be added as appendix or placed directly in the text (the statistics about questions, scale is needed).

The extension of the paper into perception of consumers should be added to the paper

The explanation why the number of participants (n=52) is so small, but the questionnaire had been available for 4 months is needed.

Author Response

Thank you for your comments. I added to the text (see word).

Reviewer 2 Report

The Introduction section needs improvement. The authors should elaborate on relevant contextual ideas and background leading to research studies and additional literature to explain why it is essential for this research study.

It is necessary to explain the research gaps in the industry that the paper seeks to close and why the paper is needed to recognize the current gaps in the literature. Before explaining the purpose of the study, please explain the research gap that this study will address. Research gap is a gap between what is known from extant literature and what has not been solved.

The weaknesses of the methods are reflected in the section Results and Discussion. The author(s) should present the discussion section. The discussions should connect the research results with relevant literature citations.

In particular, it is unclear what kind of existing issues this research focuses on. The author(s) should present the research gap in extant literature and locate the purpose of this study there.

Also, the author(s) should present the hypothesis (hypothesis are not explained in this paper). The authors should be explained in the theoretical background, and arguments from which the hypothesis was formulated. Basically, the author(s) should position the purpose of this research against the research gap. Please explain all the constructs used in the hypotheses, providing the extant literature in the theoretical background. Beside, please add an explanation of how the author(s) developed each hypothesis and from what perspective.

Author Response

(The authors gave the same response as above.)

Round 2

Reviewer 1 Report

Authors addressed all comments raised in the review form. It seems that the paper can be interested fro a potential reader.

Two more comments:

  1. check the entire manuscript for editing errors
  2. remove comment from line 408.

Reviewer 2 Report

The author responded to the requests